# Proteomic Analysis of Prehypertensive and Hypertensive Patients: Exploring the Role of the Actin Cytoskeleton

**DOI:** 10.3390/ijms25094896

**Published:** 2024-04-30

**Authors:** Sarah Al Ashmar, Gulsen Guliz Anlar, Hubert Krzyslak, Laiche Djouhri, Layla Kamareddine, Shona Pedersen, Asad Zeidan

**Affiliations:** 1Department of Basic Sciences, College of Medicine, QU Health, Qatar University, Doha 2713, Qatar; sa1903822@qu.edu.qa (S.A.A.); ga1912359@qu.edu.qa (G.G.A.); ldjouhri@qu.edu.qa (L.D.); 2Department of Clinical Biochemistry, Aalborg University Hospital, 9000 Aalborg, Denmark; hubert.krzyslak16@gmail.com; 3Biomedical Sciences Department, College of Health Sciences, QU Health, Qatar University, Doha 2713, Qatar; lkamareddine@qu.edu.qa; 4Biomedical Research Center, Qatar University, Doha 2713, Qatar

**Keywords:** hypertension, prehypertension, biomarkers, proteomics, actin cytoskeleton

## Abstract

Hypertension is a pervasive and widespread health condition that poses a significant risk factor for cardiovascular disease, which includes conditions such as heart attack, stroke, and heart failure. Despite its widespread occurrence, the exact cause of hypertension remains unknown, and the mechanisms underlying the progression from prehypertension to hypertension require further investigation. Recent proteomic studies have shown promising results in uncovering potential biomarkers related to disease development. In this study, serum proteomic data collected from Qatar Biobank were analyzed to identify altered protein expression between individuals with normal blood pressure, prehypertension, and hypertension and to elucidate the biological pathways contributing to this disease. The results revealed a cluster of proteins, including the SRC family, CAMK2B, CAMK2D, TEC, GSK3, VAV, and RAC, which were markedly upregulated in patients with hypertension compared to those with prehypertension (fold change ≥ 1.6 or ≤−1.6, area under the curve ≥ 0.8, and *q*-value < 0.05). Pathway analysis showed that the majority of these proteins play a role in actin cytoskeleton remodeling. Actin cytoskeleton reorganization affects various biological processes that contribute to the maintenance of blood pressure, including vascular tone, endothelial function, cellular signaling, inflammation, fibrosis, and mechanosensing. Therefore, the findings of this study suggest a potential novel role of actin cytoskeleton-related proteins in the progression from prehypertension to hypertension. The present study sheds light on the underlying pathological mechanisms involved in hypertension and could pave the way for new diagnostic and therapeutic approaches for the treatment of this disease.

## 1. Introduction

Hypertension affects more than one billion people worldwide, but less than half are being diagnosed and treated [1]. It is referred to as the “silent killer” because almost half of people with hypertension are unaware of their condition [1]. It affects one in three adults globally, and only 21% have it under control [2]. In the Gulf region, hypertension is one of the most important risk factors associated with coronary heart disease and stroke [3]. Based on a stepwise survey, 33% of Qatari adults have the disease [4]. Furthermore, according to WHO estimates of noncommunicable disease mortality in the GCC in 2008, 23% of deaths in Qatar are caused by cardiovascular disease [3]. Studies in Gulf countries, including Qatar, have revealed a high prevalence of hypertension (49%) among individuals with acute coronary syndrome [3].

According to the European Society of Cardiology/European Society of Hypertension (ESC/ESH), optimal blood pressure is characterized by systolic blood pressure < 120 mmHg and diastolic blood pressure < 80 mmHg. Hypertension is identified when systolic blood pressure is ≥140 mmHg and/or diastolic blood pressure is ≥90 mmHg. Elevated blood pressure levels between 120 to 139 mmHg for systolic blood pressure and/or 80 to 89 mmHg for diastolic blood pressure are classified as normal to high [5]. Studies have shown that prehypertension often progresses to hypertension and increases the risk of cardiovascular disease development [6]. Hypertension develops due to an imbalance in the physiological regulation of blood pressure, which is normally maintained by an interaction of the renal, nervous, endocrine, and cardiovascular systems [7]. It is characterized by a stiffening of the arterial wall, which forces the heart to pump blood weakly throughout the body [8]. Currently, there is no cure for hypertension. Nevertheless, blood pressure can be regulated via salt restriction, a healthy diet, weight loss, cessation of smoking, physical activity, and stress reduction, among other methods [9]. These strategies may be effective and can delay the progression of the disease in patients at high risk or patients with prehypertension. In the case of hypertension, however, these lifestyle modifications must be combined with pharmacological treatments, such as blood pressure-lowering drugs [9,10]. Although the exact cause of hypertension remains unknown, the risk of developing this disease has been strongly linked to age, unhealthy lifestyles, stress, and genetics [11]. These variables can influence the genome and proteome of an individual. Therefore, studying these changes will aid in comprehending the pathophysiology of this disease [12]. Using in vitro and in vivo models, valuable studies are being conducted to determine the potential role of specific biomarkers in hypertension; however, the methodologies needed for these studies are often costly and incompatible with large sample studies [13]. Furthermore, given the complexities of hypertension and the interconnected mechanisms that contribute to its progression, a single biomarker may be insufficient for diagnosis, staging, or prognosis. Therefore, recent advances in clinical research have focused on proteomic strategies, which entail studying the entire proteome in a cell, tissue, or organism to identify a panel of biomarkers that can provide a more accurate reflection of the disease’s proteomic profile [12]. It also permits the study of a larger population and thousands of targets simultaneously [13]. Utilizing various biological fluids, such as plasma, urine, or circulating extracellular vesicles, these proteomic approaches have resulted in numerous studies addressing the pathological mechanisms in cardiovascular diseases, such as hypertension [13]. Studies that focus on a large number of markers and their interactions, rather than a single target, can help us understand the underlying pathological pathways involved in prehypertension and hypertension and aid in the early detection of this condition.

Thus, the aim of this study was to map the altered pathways based on the differentially expressed proteins in patients with hypertension and prehypertension in comparison to healthy individuals. Using serum proteomics data, we identified a panel of proteins involved in regulating the actin cytoskeleton, which appears to play a role in the transition from prehypertension to hypertension.

## 2. Results

### 2.1. General Characteristics of the Study Population

The study included a discovery cohort of 67 participants categorized into three groups according to their blood pressure levels: a control group of 23 participants with optimal systolic blood pressure (<120 mmHg), a prehypertension group of 22 participants with systolic blood pressure between 120 and 139 mmHg, and a hypertension group of 22 participants with systolic blood pressure ≥ 140 mmHg. As shown in Table 1, the age distribution among participants revealed that individuals in the control group were younger compared to those in the disease groups. However, upon conducting post hoc analysis, no significant difference was observed in the age comparison between prehypertensive and hypertensive participants (Appendix A). Except for blood pressure levels, there were no significant variations in blood cell count, liver function, BMI, cholesterol levels, and other parameters among the three groups. In addition, post hoc analysis indicated no significant difference between the number of participants undergoing hypertension treatment (either through antihypertensive medication or specific dietary approaches) in the prehypertensive and hypertensive groups (Appendix A).

### 2.2. Distinct Proteomic Profile between Healthy, Prehypertensive, and Hypertensive Participants

The present study compared 1305 protein profiles obtained from SOMAscan analysis of plasma samples from the three groups. To visualize the data with reduced dimensionality and detect outliers, principal component analysis (PCA) was performed using protein intensity values for each sample (Appendix A). Orthogonal projections to latent structures discriminant analysis (OPLS-DA) was then used to identify clusters based on whether subjects were hypertensive, prehypertensive or not (controls). Interestingly, participants clustered based on their corresponding group along two class-discriminatory principal components (PC1 and PC2) with R2X = 0.164, R2Y = 0.546, and Q2 = 0.103 (Figure 1A). A clear separation was observed between the study groups, suggesting unique protein profiles for each group. A heatmap was also used to depict the differences in protein abundances among the three groups (Figure 1B).

Furthermore, in order to control the potential influence of medication use on group separation, a reanalysis of the same data was conducted excluding participants receiving anti-hypertensive treatment from the study cohorts. Consistently, the obtained results demonstrated clustering of participants within each respective group, confirming the reliability of our findings. OPLSDA for this analysis is represented in Appendix A.

### 2.3. Altered Protein Expression in the Control, Prehypertensive, and Hypertensive Groups

Out of the 1305 proteins detected by SOMAscan, a total of 155 proteins showed significant variation (*p*-value and *q*-value < 0.05) across the three groups (Appendix A). Volcano plots revealed 18 upregulated proteins between the hypertension and control groups (Figure 2A, Appendix A), 2 proteins upregulated in prehypertension group compared to the control group (Figure 2B, Appendix A), and 45 proteins upregulated in hypertension relative to prehypertension (Figure 2C, Appendix A). Notably, only those proteins that were differentially expressed between prehypertension and hypertension groups showed a significant *q*-value < 0.05 with a log2-fold change > 0.7 (Table 2). Interestingly, these proteins retained their significance with increased fold change in the reanalysis of the same groups excluding patients receiving hypertension medication (Appendix A). This suggests that medication use had no significant impact on our findings. As a result, we proceeded with our analysis, with all participants included.

### 2.4. Sensitivity and Specificity of the Identified Proteins between the Study Groups

To evaluate the diagnostic potential of the significant proteins, the expression of each protein across the three groups was depicted using a box plot, followed by ROC analysis. We examined the AUC of all significant proteins that were differentially expressed between the study groups to identify those with the highest diagnostic accuracy (Appendix A). An AUC value greater than or equal to 0.8 is considered an excellent diagnostic value. Among the proteins identified in the comparison between the prehypertension and hypertension groups, a total of 27 displayed an AUC ≥ 0.8, as indicated in Table 3.

### 2.5. Alterations in Proteins Related to Actin Cytoskeleton between Prehypertension and Hypertension Groups

To explore the biological processes associated with the significant proteins (*q*-value < 0.05), pathway enrichment analysis was performed using the DAVID v2023q1 bioinformatic tool. KEGG pathway analysis revealed enrichment in pathways related to PI3K-Akt signaling, MAPK signaling, pathways in cancer, RAS and RAP1 signaling, melanoma, focal adhesion, and actin cytoskeleton regulation (Figure 3A) with an enrichment score (ES) of 12.27 (Appendix A). Interestingly, most of the proteins identified in Table 3 between prehypertension and hypertension play a role in actin cytoskeleton regulation (Figure 3B). These proteins are depicted in the box plots in Figure 4.

### 2.6. Validation of the Actin Cytoskeleton Related Proteins in an Independent Cohort

To confirm our findings, we conducted external validation. New participants consisting of 25 normotensive, 24 prehypertensive, and 17 hypertensive individuals were included in the validation cohort (clinical characteristics of the participants are summarized in Appendix A). ROC analysis was performed to assess the diagnostic performance of the identified proteins in the new cohort. As shown in Figure 5 and in Table 4, the analysis revealed that the actin cytoskeleton-related proteins, including SRC, CAMK2B, CAMK2D, TEC, GSK3, VAV, RAC, GRB2, FER, TPM4, LYN, and YWHAB, exhibited an AUC > 0.75 in the validation cohort. These findings indicate an excellent ability of these proteins to discriminate between different stages of hypertension.

## 3. Discussion

Hypertension and prehypertension are common worldwide and pose significant mortality risks, contributing to major diseases such as stroke, heart attack, heart failure, and kidney damage [2]. The number of people suffering from hypertension doubled between 1990 and 2019, and nearly half of hypertensive patients are currently unaware of their condition [2]. Thus, identifying biomarkers that could facilitate the early detection of hypertension holds substantial clinical significance and could contribute to mitigating the global burden of this disease.

In this study, we utilized proteomic analysis to identify novel biomarkers that could distinguish between prehypertension and hypertension. Our results revealed a distinct variation in the proteomic profile of healthy, prehypertensive, and hypertensive participants, indicating that each disease state is characterized by a unique protein signature. Pathway analysis revealed a novel group of proteins associated with the actin cytoskeleton that effectively discriminated between the disease stages.

Vascular remodeling, which entails actin cytoskeleton reorganization, is considered a crucial pathological process in hypertension progression [14]. Although novel mediators of actin cytoskeleton dynamics are currently under investigation, their role in the progression of hypertension remains unclear. Our results identified several proteins that could distinguish between individuals with prehypertension and hypertension. The identified biomarkers were associated with pathways that are already known to contribute to hypertension, such as MAPK signaling [15], PI3K-Akt signaling [16], and RAS signaling [17]. Additionally, pathways such as focal adhesion, RAP1 signaling, and actin cytoskeleton regulation were found to be potential novel pathways implicated in the pathogenesis of hypertension. These findings align with previous quantitative proteomic analysis for hypertension by Matafora et al. [18]. Other proteomic studies have also identified different proteins and mechanisms involved in hypertension. For example, Gajjala et al. identified 27 differentially expressed molecular determinants between normotensive and hypertensive subjects [19]. Consistence with our findings, pathway analysis confirmed the involvement of the identified proteins in atherogenesis, cytoskeletal organization, and angiogenesis [19].

Our data showed upregulation of Src tyrosine family kinases, including LYN, LYNB, SRC, FYN, and C-terminal Src kinase (CSK), in hypertensive patients compared to prehypertensive patients. Some members of Src family, such as SRC, YES, and FYN, are ubiquitously expressed in various tissues and cells [20]. Src proteins are also highly expressed on vascular smooth muscle and endothelial cells [21]. Src proteins can interact with and phosphorylate growth factor receptors and integrin receptors, thereby regulating cell proliferation, adhesion, migration, and survival [22,23]. In addition, studies have demonestrated that Src proteins are critical regulators of actin cytoskeleton dynamics and focal adhesion turnover [20]. They can phosphorylate various cytoskeletal proteins, including actin, myosin, and focal adhesion proteins affecting processes such as cell migration, adhesion, and morphology [24]. Consistent with our findings, studies have shown that Src family kinases are involved in angiotensin II-induced hypertension through the phosphorylation of the myosin light chain in vascular smooth muscle cells [25]. The activation of Src family kinases has been suggested as one of the initial events in signal transduction and vascular responses induced by angiotensin II [25]. The inhibition of Src kinase was able to lower the level of blood pressure in angiotensin II-treated mice [25]. In addition, Src proteins have been implicated in the development of different cardiovascular diseases, including hypertension, coronary heart disease, ischemic heart disease, arrhythmia, and cardiomyopathy, through the modulation of cell growth, differentiation, movement, and function [21]. Therefore, Src family kinases might influence blood pressure directly via modulating the contractile machinery of blood vessels.

In addition, we observed an upregulation of CAMK2B and CAMK2D proteins among hypertensive participants in this study. These proteins belong to the calcium/calmodulin-dependent protein kinase II family that can regulate a range of substrate proteins in response to Ca^2+^ signals [26]. They are highly expressed in the neurons, cardiomyocytes, vascular smooth muscle cells, and immune cells [27,28]. In the heart, CaMKII regulates cardiomyocyte ion channels, Ca^2+^ balance, contraction, and transcription [29]. It can phosphorylate L-type calcium channels and Ryanodine Receptor (RyR), contributing to excitation–contraction coupling and cardiac contractility [28]. Accumulating evidence suggests that CAMKII contributes to cardiac remodeling and hypertrophy [30,31]. Numerous studies have reported that CaMKII plays an important role in the development of cardiac hypertrophy through the activation of impaired gene expression of atrial natriuretic peptide (ANP), brain natriuretic peptide (BNP), beta-myosin heavy chain (β-MHC), and skeletal actin [32]. Hasan et al. demonstrated that a high-salt diet resulted in left ventricular hypertrophy, cardiomyocyte hypertrophy, and myocardial fibrosis in rats, while this effect was reversed by inhibiting calcineurin and CaMKII [33]. On the other hand, CAMKII proteins can directly interact with actin filaments [34]. It is considered an upstream regulator of actin polymerization and stress fiber assembly in osteoclasts [35]. CAMK2B mediates the bundling of F-actin filaments and is associated with the actin cytoskeleton in fibroblasts [26]. These studies align with our findings, indicating that these proteins might be involved in actin cytoskeleton rearrangement leading to hypertension.

Furthermore, our results demonstrated an increase in VAV1 and RAC1 proteins in the hypertensive group compared to the prehypertensive group. VAV1 is mainly expressed in the cells of the hematopoietic system, and it is involved in biological processes such as cell signaling, cytoskeletal rearrangement, and immune responses [36]. VAV1 proteins act as guanine nucleotide exchange factors (GEF) for small G proteins of the Rho family, which plays a vital role in the regulation of the actin cytoskeleton [36]. RAC1 is an important member of the Rho GTPases [37]. It is highly expressed in the cells of the immune system, and can also be found in epithelial cells, endothelial cells, neuronal cells, and fibroblasts [37]. It is considered a cytoskeleton regulatory protein that controls cell adhesion and movement by transmitting signals to the cytoskeleton and facilitating actin remodeling [37]. VAV can activate RAC, leading to actin cytoskeleton assembly [38]. Recent evidence has indicated that the VAV-RAC pathway is involved in blood pressure regulation [39]. Additionally, RAC has been implicated in cardiac hypertrophy [40] and hypertension, possibly by affecting sodium and water balance [41]. Moreover, it has been suggested that VAV protein can interact with TEC family kinases [42]. Interestingly, in this study, we observed TEC protein upregulation in hypertensive compared to prehypertensive participants. TEC kinases are a subfamily of nonreceptor protein tyrosine kinases that are mainly involved in intracellular signaling in lymphocytes [43]. Recent reports suggest that TEC kinases also play a role in cardiovascular disease, including ischemic heart disease and cardiac hypertrophy [43]. TEC has been identified as an interactor with Rho kinase 1, a downstream kinase effector of RhoA GTPase [44]. Studies have also demonstrated the involvement of TEC kinases in actin cytoskeleton organization, with these proteins localizing to actin-rich regions of the cell [45]. Furthermore, TECs can participate in cell adhesion through their effect on VAV1 [45]. These results suggest that VAV, RAC, and TEC proteins could play a role in hypertension progression by affecting the actin cytoskeleton.

We also found significant upregulation of tropomyosin 4 (TPM4) in hypertensive compared to prehypertensive subjects. TPM4 is a member of tropomyosin family of actin-binding proteins [46]. TPM4 is primarily expressed in muscle tissues, both skeletal and cardiac, where it is involved in the regulation of muscle contraction by stabilizing actin filaments [46]. TPM4 expression has been observed in non-muscle cells, such as fibroblasts, endothelial cells, and certain types of cancer cells [47]. This protein is involved in the regulation of cytoskeleton dynamics and muscle contraction along with other sarcomeres, including actin, troponins, and tropomodulin [46]. TPM4 can interact with filamentous actin and modulate its assembly [46]. In general, tropomyosin can impact cardiac function [48]. For instance, phosphorylation of tropomyosin leads to dilated cardiomyopathy, while reducing its phosphorylation can rescue hearts affected by hypertrophic cardiomyopathy [48]. Altered tropomyosin expression has also been reported in patients with essential hypertension [49]. However, the specific relationship between TPM4 and hypertension has not yet been established. Further research is required to identify the potential role of TPM4 role in disease progression.

In this study, we found an increase in the expression of 14-3-3 protein family, which consists of conserved regulatory molecules involved in signal transduction via binding to a variety of phosphorylated proteins, such as kinases, phosphatases, and transmembrane receptors [50]. They are expressed in all eukaryotic cells and play a role in numerous regulatory pathways, including mitogenic signaling, apoptosis, and the cell cycle [50]. A study by Jia-Hua Qu et al. reported the functional role of 14-3-3 protein–protein interactions in the heart [51]. This study demonstrated that the deduced function of cardiac 14-3-3 protein–protein interactions is to control cardiac metabolic homeostasis [51]. These proteins were shown to influence actin dynamics by stabilizing cofilin, which in turn binds to actin filaments that sever in the generation of free barbed ends [52]. However, the presence of these proteins in hypertension has yet to be discovered. Therefore, our findings provide novel evidence for the potential role of this protein in hypertension progression.

Our study also showed a significant increase in GSK3 levels between prehypertensive and hypertensive participants. GSK3, which exists in two isoforms, GSK3α and GSK3β, is a serine/threonine protein kinase that regulates glycogen synthase activity [53]. GSK3 is ubiquitously expressed in many cell types, including neurons, hepatocytes, myoblasts, and adipocytes, and multiple tissues such as skeletal muscle, liver, pancreas, and adipose tissue [54]. Numerous proteins are regulated by GSK3 and, thus, it is associated with a wide range of cellular processes, such as cell proliferation, differentiation, apoptosis, the cell cycle, the immune response, and organ development [54]. GSK3 contributes to cytoskeleton dynamics by regulating Rho family proteins [53]. It can also regulate RAC1 activity [53]. Studies have shown that Src and Fyn kinases might be responsible for the regulation and phosphorylation of GSK3 at tyrosine residue [54]. The involvement of GSK3 in cardiovascular diseases is still controversial. Some studies suggested that GSK3β acts as a negative regulator of cardiac hypertrophy and is inactivated in patients with heart failure [55]. The deletion of GSK3α in mice was also reported to cause cardiac hypertrophy and contractile dysfunction [54]. Other studies indicate that the overexpression of GSK3β leads to cardiac dysfunction, and its inhibition is cardioprotective [55]. Studies using genetically modified mouse models have also shown contradictory roles of GSK3α and GSK3β in regulating cardiac homeostasis [56]. Further research is needed to examine the exact role of GSK3 in hypertension and other cardiovascular pathologies.

Finally, our study showed an increase in the expression of the adaptor protein Grb2 in hypertensive patients. GRB2 is a ubiquitously expressed adaptor protein that functions to recruit and bring together signaling molecules, enabling signal transmission [57]. It interacts with a numerous cellular proteins, including EGFR, protein tyrosine kinases, receptor tyrosine kinases, phosphatases, adaptors, and intracellular scaffolds [57]. Additionally, Grb2 is considered a key factor in actin-dependent signaling, as many of its targets contribute to actin cytoskeleton dynamics [58]. For instance, N-WASP, an important regulator of the cytoskeleton, and Caldesmon, an actin binding protein, are targets for Grb2 [58,59]. Limited research has been conducted on the function of Grb2 in hypertension. Zhang et al. demonstrated that Grb2 is involved in the response to pressure overload–cardiac hypertrophy and fibrosis [60]. Studies have also reported that Grb2 is present in rat vascular smooth muscle cells and is recruited to the plasma membrane upon stimulation with angiotensin II and mechanical stretch [61]. Proteomic analysis revealed that Grb2 is involved in myocardial damage following acute kidney injury [62]. Thus, the upregulation of Grb2 could have pathological implications by affecting the cytoskeleton in vascular or cardiac cells, potentially leading to hypertension.

### Limitations of the Study

Despite the rigorous methodology employed in this study, it is imperative to acknowledge certain limitations that might have influenced our findings. Firstly, there is an absence of specific information regarding whether hypertension is essential or secondary. As proteomic profiles may vary between these two types, we have tried excluding most chronic diseases from our study groups, reducing the possibility of having secondary hypertension. However, no exact information on this matter was provided from Qatar Biobank. Secondly, this study lacks data about the menopausal stage or polycystic ovary syndrome (PCOS) status among female participants. Since a link between menopause and hypertension has been established, the absence of this information might introduce a potential confounding factor. Thirdly, although matching for age between groups was performed, there was still a significant difference between the age of the control and disease groups. However, this difference was not significant between prehypertensive and hypertensive participants, suggesting that age does not influence the distinctions between these two groups. Furthermore, hypertension is age-related, and it is expected that control individuals, being healthy, would be younger as hypertension is less prevalent at a younger age. Finally, the validation of the identified actin-related proteins by Western blotting might be important in such proteomics studies. However, we addressed this limitation by conducting external validation using an independent cohort, thus confirming our findings.

## 4. Materials and Methods

### 4.1. Study Participants

This study utilized data from the Qatar Biobank, a national population-based cohort study launched in December 2012 (https://www.qatarbiobank.org.qa/, accessed on 6 March 2023) [63]. A total of 133 participants, divided between discovery and validation cohorts, were included in this study. The discovery cohort, involved in the initial identification of the biomarkers, comprised 22 hypertensive patients (blood pressure ≥ 140 mmHg), 22 prehypertensive patients (blood pressure 120–139 mmHg), and a control group of 23 normotensive donors (blood pressure < 120 mmHg). The validation cohort, which served to independently confirm the significance of the identified biomarkers, consisted of 25 controls, 24 prehypertensive patients, and 17 hypertensive participants. The classification of the groups followed the European Society of Cardiology/European Society of Hypertension (ESC/ESH) guidelines [5] and the WHO definition of hypertension (140/90 mmHg or higher) [2]. Exclusion criteria encompassed individuals with diabetes, obesity, moderate to mild anemia, thrombocytopenia, high cholesterol, hypo- or hyperthyroidism, autoimmune diseases, angina, heart attack, cancer, or liver disease. All participants provided informed consent, and the study was approved by the QBB institutional review board (E-2021-QF-QBB-RES-ACC-00021-0160). The proteomics and clinical data for the study participants are provided in Appendix A.

### 4.2. Samples Collection and Processing

Each participant provided a blood sample of approximately 60 mL after fasting overnight. The samples were transported to the laboratory at Hamad Medical Center in Doha for biochemical and hematological assessments. The EDTA blood samples were then subjected to centrifugation to isolate the different blood components, including plasma, buffy coat, and erythrocytes. Portions of the samples were also divided into aliquots and preserved either in liquid nitrogen for long-term storage or at −80 °C for future analysis [63].

### 4.3. Physical and Clinical Analysis

A Seca stadiometer and Seca Bio Impedance Analysis (Seca GmbH & Co. KG, Hamburg, Germany) were used to measure the height, weight, waist, and hip of each participant (Anthropometry). Body Mass Index (weight in kg/height in m^2^) and waste-to-hip ratio (waist/hip in cm) were then calculated. Blood pressure was measured using an Omron 705 automated instrument (Omron Corporation, Kyoto, Japan). An average of two or three (if the first two measurements differed by 5 mmHg) diastolic and systolic blood pressure measurements were performed. Clinical biomarkers, including complete blood count, lipid profile, HbA1c%, liver function tests, and C-reactive protein, were analyzed as previously described [63,64].

### 4.4. Analysis Multiplexed Aptamer-Based (SOMAscan) Proteomic Platform

Analysis was performed utilizing a Multiplexed aptamer-based (SOMAscan) proteomic platform (v3.1, Somalogic, Boulder, CO, USA) at Weill Cornell Medicine—Qatar. The aptamer-based SOMAscan assay, a relatively new addition to the targeted affinity proteomics field, enables the simultaneous measurement and quantitation of over 1000 proteins using unique SOMAmers. The assay was developed for the analysis of clinical samples and is distinguished by its high complexity and dynamic range [65]. As previously described [66,67], EDTA-plasma samples were incubated with epitope-specific aptamers (SOMAmers) coupled to beads. The proteins bound to the beads were biotinylated, and the resulting complexes of biotinylated target proteins and fluorescence-tagged SOMAmers were then subjected to photocleavage and recaptured on streptavidin beads and subsequently eluted and quantified by hybridization to custom arrays of SOMAmer-complementary oligonucleotides. Different standards were used as references to process the resulting raw intensities. Data for 1305 aptamers were obtained. There were no excluded data points, and repeated measurements for two QC samples were used for quality control. For protein annotations, aptamer IDs were used as primary identifiers. The identifiers were associated with the proteins targeted by the corresponding aptamers using UniProt and Entrez gene names.

### 4.5. Statistical Analysis

Demographics and clinical data were analyzed using IBM SPSS Statistics 29.0.0.0 (SPSS, Chicago, IL, USA) and displayed as mean with standard deviation (mean ± SD). Correction for age, sex, and BMI has been performed to minimize any confounding effects. The differences between the three study groups were examined using one-way ANOVA (analysis of variance) for categorical variables and the Chi-square test for nominal variables, followed by Dunnett’s post hoc test. Unsupervised principal component analysis (PCA) was performed, using log-transformed protein intensity values (RFUs) to evaluate data inclinations, followed by supervised orthogonal projections to latent structures discriminant analysis (OPLS-DA) to identify proteins associated with sample grouping using SIMCA^®^ 16.0.1. (Sartorius, Goettingen, Germany). After PCA, two participants from the control group and one participant from the prehypertension group were considered outliers and excluded from the study. The RFU values of identified proteins were uploaded to Perseus 2.0.7.0 (Max Planck Institute of Biochemistry, Martinsried, Germany) [68] and subsequently log_2_ transformed. Histograms were used to assess data distribution. The differentially expressed proteins in healthy and pre- and hypertensive individuals were identified using one-way ANOVA. *p*-value adjustment was performed using the Benjamini–Hochberg method, and *q*-values were obtained through False Discovery Rate (FDR) estimation. Proteins were considered statistically significant if *q*-value < 0.05 and log_2_-fold change (FC) ≥ 0.7 or ≤−0.7 and were depicted using volcano plots in GraphPad Prism 9.1.2. (GraphPad Software, La Jolla, CA, USA). Heatmaps were also generated in Perseus. KEGG (Kyoto Encyclopedia of Genes and Genomes) pathway analysis was performed on significantly expressed proteins by the Database for Annotation, Visualization, and Integrated Discovery (DAVID) Bioinformatics Resources 6.8 (https://david.abcc.ncifcrf.gov, accessed on 6 March 2023) [69,70]. Enrichment scores (ESs) with Benjamini–Hochberg-corrected *p*-values were shown. Protein–protein interactions were established using STRINGDB version 11.5 (https://string-db.org, accessed on 6 March 2023) [71]. Important proteins distinguishing prehypertensive and hypertensive patients from healthy individuals were selected and depicted as box plots and presented through receiver operating characteristic (ROC) analysis using raw RFU values.

## 5. Conclusions

In recent years, proteomics research has expanded significantly with the aim of identifying biological markers associated with various diseases, including hypertension. In our study, we found 45 proteins that exhibited significant upregulation in hypertensive patients compared to those with prehypertension. Pathway analysis further elucidated a subset of these proteins, including the SRC family, CAMK2B, CAMK2D, TEC, GSK3, VAV, and RAC, which play a role in actin cytoskeleton arrangement. These proteins demonstrated strong diagnostic performance in both discovery and validation groups. This exploratory study contributes to the global effort to identify distinct protein biomarkers that can differentiate between prehypertensive and hypertensive individuals. However, further research is necessary to validate and confirm these findings before their implementation in clinical practice. This research holds the potential to open new avenues for diagnosing and managing hypertension.

## Figures and Tables

**Figure 1 ijms-25-04896-f001:**
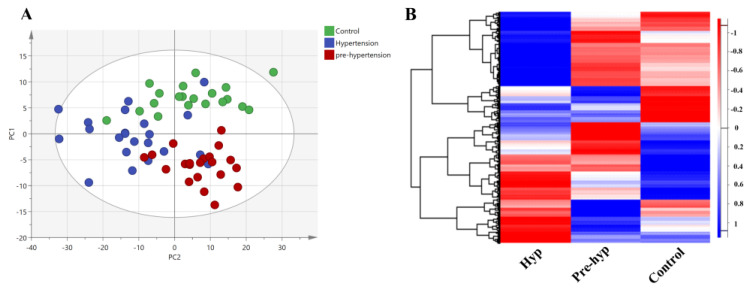
Proteomic analysis of the groups. (**A**) Orthogonal partial least squares discriminant analysis (OPLS-DA) scatter plot showing the separation between the three groups based on their proteomic signature. (**B**) Heatmap of all 1305 proteins showing an unsupervised hierarchical clustering of altered proteins between the three groups. PC1: principal component 1; PC2: principal component.

**Figure 2 ijms-25-04896-f002:**
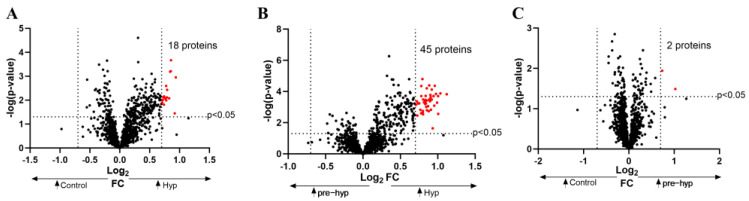
Volcano plots. (**A**) Volcano plot representing differentially expressed proteins between control and hypertension groups. (**B**) Volcano plot representing differentially expressed proteins between control and prehypertension groups. (**C**) Volcano plot representing differentially expressed proteins between hypertension and prehypertension groups.

**Figure 3 ijms-25-04896-f003:**
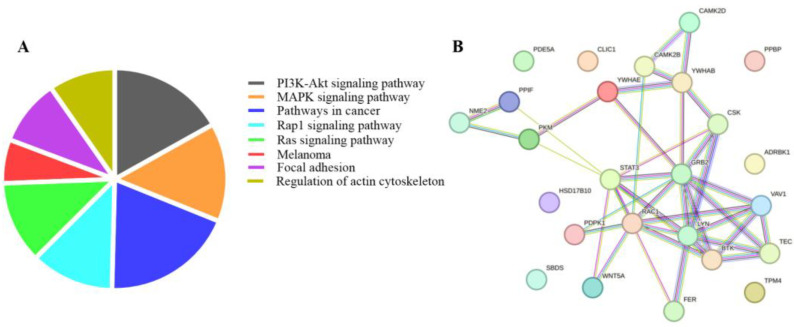
(**A**) KEGG pathway analysis of the significant proteins resulting from ANOVA. Cluster 1 (enrichment score = 12.27) is represented in a pie chart in terms of the number of proteins mapped per pathway. (**B**) Network analysis of the significant proteins between prehypertension and hypertension with AUC ≥ 0.8. Number of nodes: 25; number of edges: 54; average node degree: 4.32; average local clustering coefficient: 0.482; expected number of edges: 17; protein–protein interaction enrichment *p*-value: 1.15 × 10^−12^.

**Figure 4 ijms-25-04896-f004:**
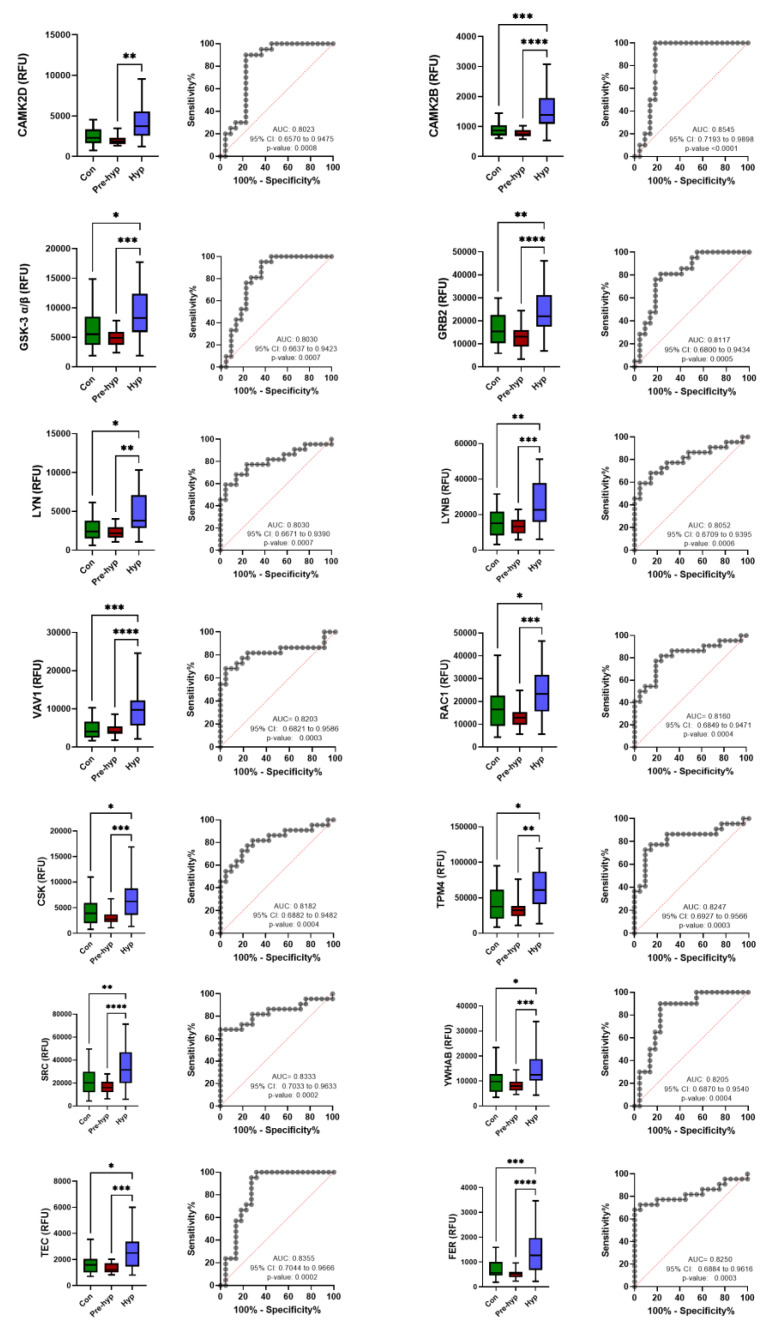
Boxplots and ROC curves of actin cytoskeleton-related proteins significantly different between prehypertension and hypertension groups. The relative fluorescence unit (RFU) was used for boxplots. *AUC*: Area Under Curve, *CI*: Confidence Interval; *CAMK2B:* Calcium/calmodulin-dependent protein kinase type II subunit beta; *CAMK2D:* Calcium/calmodulin-dependent protein kinase type II subunit delta; *GSK-3 α/β:* Glycogen synthase kinase-3 alpha/beta; *GRB2:* Growth factor receptor-bound protein 2; *LYN:* Tyrosine-protein kinase Lyn; *LYNB:* Tyrosine-protein kinase Lyn isoform B; *VAV1:* Proto-oncogene vav; *RAC1:* Ras-related C3 botulinum toxin substrate 1; *CSK:* Tyrosine-protein kinase CSK; *TPM4:* Tropomyosin alpha-4 chain; *SRC:* Proto-oncogene tyrosine-protein kinase Src; *YWHAB:* 14-3-3 protein beta/alpha; *TEC:* Tyrosine-protein kinase Tec; *FER:* Tyrosine-protein kinase Fer. * *p* < 0.05, ** *p* < 0.01, *** *p* < 0.001, **** *p* < 0.0001.

**Figure 5 ijms-25-04896-f005:**
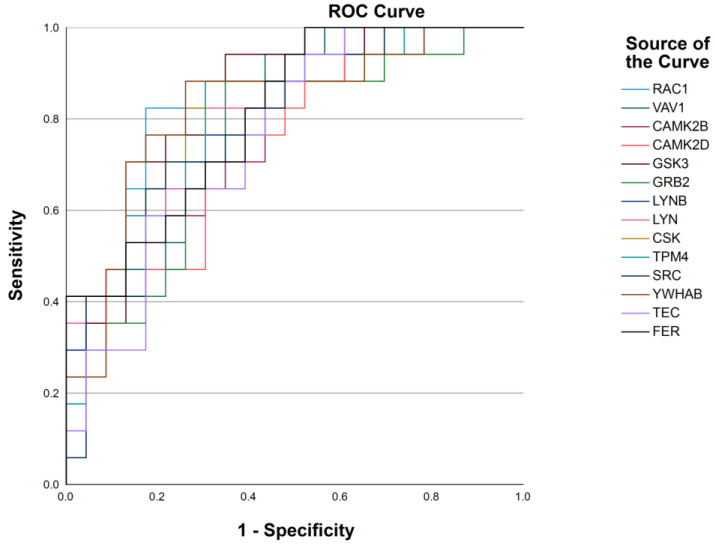
ROC analysis of the actin cytoskeleton-related proteins significantly different between prehypertension and hypertension in the validation cohort.

**Table 1 ijms-25-04896-t001:** Clinical characteristics of the study participants in the discovery cohort. Data are represented as mean ± standard deviation. One-way ANOVA was used to calculate the *p*-value.

	Controls	Prehypertension	Hypertension	*p*-Value
Number of participants	23	22	22	
Gender (M/F)	11/12	12/10	12/10	0.99
Age	45.2 ± 4.3	51.7 ± 5.4	50.2 ± 9.1	0.003
BMI	26 ± 3.1	26.8 ± 2	26.92 ± 2.3	0.39
BMI (Males)	27.03 ± 2.1	26.86 ± 2	26.73 ± 2.4	0.94
BMI (Females)	24.8 ± 3.7	26.7 ± 2.1	27.2 ± 2.2	0.15
Hemoglobin g/dL	13.3 ± 1.3	14.3 ± 1.6	14.1 ± 1.6	0.077
Red Blood Cell × 10^6^/µL	4.8 ± 0.6	5 ± 0.6	5 ± 0.5	0.46
White Blood Cell × 10^3^/µL	6.01 ± 1.3	6.9 ± 1.9	5.9 ± 1.2	0.072
Cholesterol Total mmol/L	5.04 ± 0.6	4.9 ± 0.7	5.12 ± 0.7	0.54
LDL-Cholesterol mmol/L	2.9 ± 0.8	2.8 ± 0.6	3.12 ± 0.64	0.33
HbA1c%	5.4 ± 0.3	5.4 ± 0.4	5.6 ± 0.4	0.24
Systolic blood pressure	105 ± 8.4	127.4 ± 5.6	150 ± 12.4	<0.001
Diastolic blood pressure	68.5 ± 6	83.2 ± 6.4	91.9 ± 9.8	<0.001
Albumin g/L	45.9 ± 1.3	46.3 ± 1.9	46.4 ± 2.8	0.67
ALT U/L	26.9 ± 21	25.5 ± 9.7	27.2 ± 15.1	0.93
AST U/L	21.2 ± 6.6	20.6 ± 5.8	23 ± 9.7	0.55
C-Reactive Protein mg/L	5.05 ± 0.2	5.55 ± 2	5.4 ± 1.4	0.54
Hypertension treatment				
Tablets	0	2	5	0.043
DASH	0	2	5	0.043

LDL: Low Density Lipoprotein. BMI: Body Mass Index. HbA1c: Hemoglobin A 1C. ALT: Alanine Amino Transferase. AST: Aspartate Amino Transferase. DASH: Dietary Approaches to Stop Hypertension (a diet rich in potassium, calcium, magnesium, fiber and protein, low in saturated fat and salt).

**Table 2 ijms-25-04896-t002:** Significantly differentially expressed proteins between prehypertension and hypertension groups. *p*-value and *q*-value < 0.05; −0.7 < log2 FC > 0.7. FC: fold change.

Hypertension|Prehypertension					
UniProt	Gene	Protein Name	Log2FC	*p*-Value	*q*-Value
P16591	*FER*	Tyrosine-protein kinase Fer	1.12	<0.001	<0.001
Q06187	*BTK*	Tyrosine-protein kinase BTK	1.03	<0.001	<0.001
Q08752	*PPID*	Peptidyl-prolyl cis-trans isomerase D	1.01	<0.001	0.02
P41240	*CSK*	Tyrosine-protein kinase CSK	1.00	<0.001	<0.001
P05771	*PRKCB*	Protein kinase C beta type (splice variant beta-II)	0.99	<0.001	0.01
P15498	*VAV1*	Proto-oncogene vav	0.98	<0.001	<0.001
Q99714	*HSD17B10*	3-hydroxyacyl-CoA dehydrogenase type-2	0.96	<0.001	<0.001
P30405	*PPIF*	Peptidyl-prolyl cis-trans isomerase F; mitochondrial	0.95	<0.001	0.01
P12931	*SRC*	Proto-oncogene tyrosine-protein kinase Src	0.94	<0.001	<0.001
P18669	*PGAM1*	Phosphoglycerate mutase 1	0.93	<0.001	0.11
Q04759	*PRKCQ*	Protein kinase C theta type	0.92	<0.001	<0.001
P14618	*PKM2*	Pyruvate kinase PKM	0.92	<0.001	0.01
P17252	*PRKCA*	Protein kinase C alpha type	0.91	<0.001	0.01
O76074	*PDE5A*	cGMP-specific 3′;5′-cyclic phosphodiesterase	0.90	<0.001	<0.001
P67936	*TPM4*	Tropomyosin alpha-4 chain	0.90	<0.001	<0.001
P07948	*LYNB*	Tyrosine-protein kinase Lyn; isoform B	0.90	<0.001	0.01
Q15796	*SMAD2*	Mothers against decapentaplegic homolog 2	0.89	<0.001	0.01
P31946	*YWHAB*	14-3-3 protein family	0.89	<0.001	<0.001
O15530	*PDPK1*	3-phosphoinositide-dependent protein kinase 1	0.88	<0.001	0.01
Q9NQU5	*PAK6*	Serine/threonine-protein kinase PAK 6	0.88	<0.001	0.01
Q9NYA1	*SPHK1*	Sphingosine kinase 1	0.87	<0.001	0.01
P06241	*FYN*	Tyrosine-protein kinase Fyn	0.86	<0.001	0.01
P42680	*TEC*	Tyrosine-protein kinase Tec	0.85	<0.001	<0.001
P07948	*LYN*	Tyrosine-protein kinase Lyn	0.85	<0.001	0.01
P42574	*CASP3*	Caspase-3	0.85	<0.001	0.01
O95219	*SNX4*	Sorting nexin-4	0.84	<0.001	0.01
P62993	*GRB2*	Growth factor receptor-bound protein 2	0.84	<0.001	<0.001
Q9Y3A5	*SBDS*	Ribosome maturation protein SBDS	0.84	<0.001	0.01
P78344	*EIF4G2*	Eukaryotic translation initiation factor 4 gamma 2	0.83	<0.001	0.01
Q13557	*CAMK2D*	Calcium/calmodulin-dependent protein kinase type II subunit delta	0.83	<0.001	0.01
P22392	*NME2*	Nucleoside diphosphate kinase B	0.82	<0.001	0.01
O43488	*AKR7A2*	Aflatoxin B1 aldehyde reductase member 2	0.81	<0.001	0.01
Q15056	*EIF4H*	Eukaryotic translation initiation factor 4H	0.80	<0.001	0.02
Q8N1Q1	*CA13*	Carbonic anhydrase 13	0.80	<0.001	0.01
P10809	*HSPD1*	60 kDa heat shock protein; mitochondrial	0.80	<0.001	0.02
Q13554	*CAMK2B*	Calcium/calmodulin-dependent protein kinase type II subunit beta	0.79	<0.001	<0.001
O00299	*CLIC1*	Chloride intracellular channel protein 1	0.78	<0.001	<0.001
P25098	*ADRBK1*	beta-adrenergic receptor kinase 1	0.78	<0.001	<0.001
P54646	*PRKAA2/B2/G1*	AMP Kinase (alpha2beta2gamma1)	0.78	<0.001	0.02
Q9NP97	*DYNLRB1*	Dynein light chain roadblock-type 1	0.77	<0.001	0.01
P49840	*GSK3A/B*	Glycogen synthase kinase-3 alpha/beta	0.75	<0.001	0.01
P63000	*RAC1*	Ras-related C3 botulinum toxin substrate 1	0.73	<0.001	0.01
P02775	*PPBP*	Connective tissue-activating peptide III	0.73	<0.001	0.02
P40763	*STAT3*	Signal transducer and activator of transcription 3	0.72	<0.001	0.01
P31946	*YWHAB*	14-3-3 protein beta/alpha	0.71	<0.001	0.01

**Table 3 ijms-25-04896-t003:** Sensitivity and specificity based on ROC analysis of proteins with AUC ≥ 0.8, *p* and *q*-value < 0.05. AUC: Area Under Curve; CI: Confidence Interval.

Hypertension|Prehypertension							
Gene	Protein Name	AUC	95% CI	*p*-Value	*q*-Value	Sensitivity%	Specificity%
*CAMK2D*	Calcium/calmodulin-dependent protein kinase type II subunit delta	0.80	0.65 to 0.94	0.0008	0.0051	90	77.27
*GSK3A/B*	Glycogen synthase kinase-3 alpha/beta	0.80	0.66 to 0.94	0.0007	0.0059	80.95	72.73
*LYN*	Tyrosine-protein kinase Lyn	0.80	0.66 to 0.93	0.0007	0.0054	77.27	76.19
*PPIF*	Peptidyl-prolyl cis-trans isomerase F	0.80	0.67 to 0.93	0.0007	0.0065	72.73	76.19
*PPBP*	Connective tissue-activating peptide III	0.80	0.67 to 0.93	0.0007	0.019	75	77.27
*LYNB*	Tyrosine-protein kinase Lyn; isoform B	0.80	0.67 to 0.93	0.0006	0.0054	72.73	76.19
*GRB2*	Growth factor receptor-bound protein 2	0.81	0.68 to 0.94	0.0005	0.0044	80.95	77.27
*NME2*	Nucleoside diphosphate kinase B	0.81	0.67 to 0.94	0.0004	0.0057	85.71	72.73
*RAC1*	Ras-related C3 botulinum toxin substrate 1	0.81	0.68 to 0.94	0.0004	0.0058	81.82	76.19
*STAT3*	Signal transducer and activator of transcription 3	0.81	0.67 to 0.95	0.0006	0.0054	70	80
*CSK*	Tyrosine-protein kinase CSK	0.81	0.68 to 0.94	0.0004	0.0043	81.82	71.43
*ADRBK1*	beta-adrenergic receptor kinase 1	0.81	0.68 to 0.95	0.0004	0.0044	80.95	77.27
*VAV1*	Proto-oncogene vav	0.82	0.68 to 0.95	0.0003	0.0044	81.82	76.19
*YWHAB*	14-3-3 protein beta/alpha	0.82	0.68 to 0.95	0.0004	0.0061	90	77.27
*TPM4*	Tropomyosin alpha-4 chain	0.82	0.69 to 0.95	0.0003	0.0047	77.27	85.71
*FER*	Tyrosine-protein kinase Fer	0.82	0.68 to 0.96	0.0003	0.0045	72.73	95
*SRC*	Proto-oncogene tyrosine-protein kinase Src	0.83	0.70 to 0.96	0.0002	0.0041	72.73	80.95
*PKM2*	Pyruvate kinase PKM	0.83	0.70 to 0.96	0.0002	0.005	81.82	80
*TEC*	Tyrosine-protein kinase Tec	0.83	0.70 to 0.96	0.0002	0.002	95.24	72.73
*PDE5A*	cGMP-specific 3′;5′-cyclic phosphodiesterase	0.83	0.71 to 0.96	0.0002	0.0042	80.95	77.27
*CLIC1*	Chloride intracellular channel protein 1	0.83	0.71 to 0.96	0.0001	0.00072	90.48	77.27
*PRKCQ*	Protein kinase C theta type	0.84	0.71 to 0.97	0.0002	0.0025	80.95	85
*PDPk1*	3-phosphoinositide-dependent protein kinase 1	0.84	0.72 to 0.96	0.0002	0.005	72.73	84.21
*BTK*	Tyrosine-protein kinase BTK	0.84	0.72 to 0.97	0.0001	0.0046	81.82	85
*CAMK2B*	Calcium/calmodulin-dependent protein kinase type II subunit beta	0.85	0.71 to 0.98	<0.0001	0.0013	100	81.82
*SBDS*	Ribosome maturation protein SBDS	0.85	0.74 to 0.97	<0.0001	0.0052	80.95	76.19
*HSD17B10*	3-hydroxyacyl-CoA dehydrogenase type-2	0.86	0.76 to 0.97	<0.0001	0.0008	76.19	80

**Table 4 ijms-25-04896-t004:** Comparison between AUC of the actin cytoskeleton related proteins following ROC analysis conducted between prehypertension and hypertension in the discovery and validation groups. AUC: Area Under Curve.

Gene	Protein Name	Validation Cohort		Discovery Cohort	
		AUC	*p*-Value	AUC	*p*-Value
*RAC1*	Ras-related C3 botulinum toxin sub-strate 1	0.83	0.0003	0.81	0.0004
*VAV1*	Proto-oncogene vav	0.79	0.001	0.82	0.0003
*CAMK2B*	Calcium/calmodulin-dependent protein kinase type II subunit beta	0.78	0.003	0.85	<0.0001
*CAMK2D*	Calcium/calmodulin-dependent protein kinase type II subunit delta	0.75	0.007	0.80	0.0008
*GSK3*	Glycogen synthase kinase-3	0.83	0.0003	0.80	0.0007
*GRB2*	Growth factor receptor-bound protein 2	0.75	0.008	0.81	0.0005
*LYNB*	Tyrosine-protein kinase Lyn; isoform B	0.81	0.001	0.80	0.0006
*LYN*	Tyrosine-protein kinase Lyn	0.82	0.001	0.80	0.0007
*CSK*	Tyrosine-protein kinase CSK	0.84	0.0002	0.81	0.0004
*TPM4*	Tropomyosin alpha-4 chain	0.81	0.001	0.82	0.0003
*SRC*	Proto-oncogene tyrosine-protein kinase Src	0.79	0.002	0.83	0.0002
*YWHAB*	14-3-3 protein beta/alpha	0.82	0.001	0.82	0.0004
*TEC*	Tyrosine-protein kinase Tec	0.76	0.005	0.83	0.0002
*FER*	Tyrosine-protein kinase Fer	0.81	0.001	0.82	0.0003

## Data Availability

The data used in this study are available in Appendix A.

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
