# Peer review of "Proteomic Analysis of Prehypertensive and Hypertensive Patients: Exploring the Role of the Actin Cytoskeleton"

_ijms, 2024, doi:10.3390/ijms25094896_

Round 1

Reviewer 1 Report

Comments and Suggestions for Authors

The study entitled “Proteomic analysis of prehypertensive and hypertensive patients: Exploring the role of actin cytoskeleton” by Sarah Al Ashmar et al  pretend to contributes to identify distinct protein expression that may function as biomarkers to identify between prehypertensive and hypertensive individuals. The study is performed within the Qatari population using Slow off-rate modified aptamer proteomics which is a new attractive analytical tool for rapid proteomic assays in prehypertensive and hypertensive patients: There is no doubt that the study contributes to a bets knowledge of the disease however I have some concerns:

1. It is clear that the use of PCA is to know the relationships between the variables and among the variables and observe trends, jumps, clusters and outliers. However, it is not clear which variables were analized: those from Table 1 or the fold change protein expression?. It must be clear so that anyone can understand it, even if they are not an expert in the statistical method used.

2.  The authors claim several proteins as Biomarker candidates; however, these statements are quite adventurous since there may be different conditions that can occur with the overexpression of the aforementioned proteins.

3.  The novelty of the study is related to actin cytoskeleton regulating proteins described, however, the analyzed samples are plasma and it is difficult to establish the origin of the overexpressing proteins detected in the study. It would be interesting if the authors discussed the action and target of up-regulated proteins such as Grb2 that have an intracellular localization. It is also important that the authors discussed the impact that some proteins have on target organs that turn out to be affected during hypertension.

Authors considered:  a cytoskeleton regulatory protein that controls cell adhesion and movement by transmitting signals to the cytoskeleton and facilitating actin remodeling, in what cells this protein may impact? Please discuss.

4. Authors did not mention if hypertensive patients were receiving treatment,  the kind of treatment and if treatment could be affecting the study.

5. It is important that some of the up-regulated proteins that authors claim to impact on actin-cytoskeleton might be validated by western-blot.

6. Conclusions must also highlight their findings

Minor points:

It is important to put into context about where the proteomics is being performed. The abstract and final part of the introduction, where the study is described, it does not mention the biological system used in the study. 

Author Response

Responses to Reviewer 1 comments

Thank you for your constructive input on our manuscript and we appreciate the time you dedicated to reviewing our work. Please find the detailed responses below and the corresponding revisions in track changes in the re-submitted files.

  1. It is clear that the use of PCA is to know the relationships between the variables and among the variables and observe trends, jumps, clusters and outliers. However, it is not clear which variables were analized: those from Table 1 or the fold change protein expression?. It must be clear so that anyone can understand it, even if they are not an expert in the statistical method used.

 Response: Thank you for your insightful feedback. To clarify, PCA was conducted as an unsupervised analysis using the protein expression data of all identified proteins. We acknowledge the oversight in specifying the variables used for PCA in our previous manuscript. To rectify this, we have incorporated the statement "PCA was performed using protein intensity values for each sample" into both the Results (section 2.2, line 123) and Methods (section 4.5, line 504).

  1. The authors claim several proteins as Biomarker candidates; however, these statements are quite adventurous since there may be different conditions that can occur with the overexpression of the aforementioned proteins.

 Response: Thank you for your constructive feedback. In response to your suggestion, we have removed the terms "biomarker candidates" and "potential biomarkers" from the manuscript to ensure a more accurate representation of our findings and to avoid overstating the implications of our results.

  1. The novelty of the study is related to actin cytoskeleton regulating proteins described, however, the analyzed samples are plasma and it is difficult to establish the origin of the overexpressing proteins detected in the study. It would be interesting if the authors discussed the action and target of up-regulated proteins such as Grb2 that have an intracellular localization. It is also important that the authors discussed the impact that some proteins have on target organs that turn out to be affected during hypertension.

Authors considered:  a cytoskeleton regulatory protein that controls cell adhesion and movement by transmitting signals to the cytoskeleton and facilitating actin remodeling, in what cells this protein may impact? Please discuss.

 Response: Thank you for your valuable input. We have duly noted your suggestion and incorporated it into the discussion section of our manuscript. We provided detailed information for each protein analyzed, including its cellular expression, functional actions, and targets within the context of our study. We appreciate your contribution to improving the comprehensiveness of our study.

  1. Authors did not mention if hypertensive patients were receiving treatment, the kind of treatment and if treatment could be affecting the study.

Response: Thank you for bringing up this important point. We have addressed your concern by including data regarding the treatment of hypertensive patients (in Table 1) and its potential impact on our study. Chi-square and post hoc analyses revealed no significant difference in the number of patients receiving treatment between the prehypertensive and hypertensive groups (p-value=0.41), as detailed in the Results section (line 103) and Supplementary Table S1. Additionally, we conducted further analysis excluding participants taking medication. Interestingly, the results from this analysis demonstrated that our proteins of interest still exhibited significant differences between the groups, suggesting that medication does not influence the proteomic results. These findings have been incorporated into the Results section (line 132 and line 152), supplementary Figure S2 and Table S4.

  1. It is important that some of the up-regulated proteins that authors claim to impact on actin-cytoskeleton might be validated by western-blot.

Response: Thank you for your suggestion. While we acknowledge the importance of validation, our study was primarily designed as an initial discovery-based investigation aimed at identifying potential proteins capable of discriminating between our study groups. We recognize the value of additional validation to strengthen the conclusions of our study. However, obtaining ethical approval for the retrieval of serum samples from the Qatar Biobank, as required for Western blot, would be a time-consuming process and was not within timeframe of this study. We highlighted this limitation in the discussion section of the manuscript (line 437). As an alternative approach, we have conducted external validation by including new participants from the Qatar Biobank in an independent validation cohort. ROC analysis was performed for the actin cytoskeleton-related proteins in this validation group. Interestingly, the results of this analysis demonstrated high significance and excellent diagnostic ability (AUC > 0.75) of the identified proteins. This new analysis has been added to the results section of the manuscript (line 202, Figure 5, and Table 4).

  1. Conclusions must also highlight their findings

Response: Thank you for your comment. We have revised the conclusion to align with your suggestion, as indicated in line 529 of the manuscript. 

Minor points:

It is important to put into context about where the proteomics is being performed. The abstract and final part of the introduction, where the study is described, it does not mention the biological system used in the study. 

Response: In response to your suggestion, we have included the information that serum samples were used to obtain the proteomics data in both abstract and the final part of the introduction to ensure that readers have a clear understanding of the biological system under investigation.

Reviewer 2 Report

Comments and Suggestions for Authors

Proteomic profiling  using SOMAscan proteomic platform was conducted on Qatari individuals with prehypertension, hypertension, and normal blood pressure. The results revealed a cluster of proteins, including the SRC family, CAMK2B and CAMK2D, TEC, GSK3, VAV, and RAC, that were markedly upregulated in patients with hypertension compared to those with prehypertension. Pathway analysis showed that most of these proteins play a role in actin cytoskeleton remodeling. Actin cytoskeleton reorganization affects various biological processes that contribute to the maintenance of blood pressure, including vascular tone, endothelial function, cellular signaling, inflammation, fibrosis, and mechanosensing. Therefore, the findings of this study suggest a potential novel role of actin cytoskeleton related proteins in the progression from prehypertension to hypertension. The dynamic rearrangement of the actin cytoskeleton is key in maintaining vascular tone and plasticity specifically important in the regulation of vascular diameter related to pressure dependent myogenic ton in hypertension.

comments

-I would avoid writing that this is the first study done on the Qatari population unless this population has genetic characteristics that distinguish it from other populations. Explain the differences between the study of the Qatari population and the studies performed on other hypertensive populations in atria. countries and whether the results of the study are like those in the literature.

-Explain the criteria for the diagnosis of prehypertension and hypertension The limit of 130 mm Systolic and 90 diastolic defines hypertensive patients from non-hypertensive patients. For example, table 1 mean values ​​of 127.4 ± 5.6 Some of these patients could have hypertension.

-The cause of hypertension (essential or secondary) is not reported.

-Explain the selection criteria for hypertensive patients, those with prehypertension and controls. Out of 60,000 participants, approximately 67 subjects of very variable ages (30-68 years) were recruited. A portion of the women were in menopause which may change proteomics.

-actin cytoskeleton-regulating proteins are influenced by age and this needs to be discussed

-Explain how the controls were chosen, because the age was different in the three groups and report how many female patients were in menopause, how many had PCOS which is associated with alterations of the proteome and often hypertension.

-A point to clarify is that the authors exclude patients with obesity but in the table many patients were overweight and obese considering the average BMI reported. Also explain the impact of BMI on the assessment. Report BMI in males and females. Report how many women used contraceptives.

-An important point to report is whether patients in the three groups were on any diet that could influence proteomics

-Explain whether hypertension therapy can influence proteomics and discuss whether each drug may have changed proteomics. Specify whether hypertension therapy and each individual drug can influence the results. This is to explain whether the difference between prehypertension and hypertension is linked to the effect of therapy on proteomics. In figure 4 the standard deviation in cases with hypertension is very high and the authors must discuss whether this is linked to the different types of therapy used

Author Response

Responses to reviewer 2 comments

Thank you for your constructive input on our manuscript and we appreciate the time you dedicated to reviewing our work. Please find the detailed responses below and the corresponding revisions in track changes in the re-submitted files.

-I would avoid writing that this is the first study done on the Qatari population unless this population has genetic characteristics that distinguish it from other populations. Explain the differences between the study of the Qatari population and the studies performed on other hypertensive populations in atria. countries and whether the results of the study are like those in the literature.

Response: Thank you for your suggestion. We have removed the sentence "first study done on the Qatari population" as the study aims to identify biomarkers associated with hypertension in general, rather than limiting it to Qatari population. We have also supplemented the discussion with examples of other proteomics studies on hypertension, providing a comparative analysis with our findings (discussion section, line 249)

-Explain the criteria for the diagnosis of prehypertension and hypertension The limit of 130 mm Systolic and 90 diastolic defines hypertensive patients from non-hypertensive patients. For example, table 1 mean values â€‹â€‹of 127.4 ± 5.6 Some of these patients could have hypertension.

Response: Thank you for your valuable comment. In our study, we have used specific criteria for the diagnosis of prehypertension and hypertension. Patients were categorized based on their blood pressure readings as follows: normal blood pressure (<120 mmHg), prehypertension (120-139 mmHg), and hypertension (≥140 mmHg for systolic and/or ≥90 mmHg for diastolic). We have now clarified in our manuscript that our classification aligns with the guidelines provided by the European Society of Cardiology/European Society of Hypertension (ESC/ESH) and the World Health Organization (WHO), which define hypertension as blood pressure of 140/90 mmHg or higher (methods section, line 452). We appreciate your input and have ensured that the criteria for hypertension diagnosis are clearly outlined in our study.

-The cause of hypertension (essential or secondary) is not reported.

Response: Thank you for your insightful comment. We acknowledge the absence of specific information regarding whether hypertension was essential or secondary. Although we excluded most other chronic diseases from our study groups, reducing the possibility of having secondary hypertension, no exact information on this matter was provided from Qatar Biobank. Therefore, we have included this limitation in the discussion section of our study (line 423).

-Explain the selection criteria for hypertensive patients, those with prehypertension and controls. Out of 60,000 participants, approximately 67 subjects of very variable ages (30-68 years) were recruited. A portion of the women were in menopause which may change proteomics.

Response: Thank you for bringing attention to this matter. Our study employed stringent inclusion criteria to ensure the selection of participants without various comorbidities such as diabetes, obesity, anemia, thrombocytopenia, high cholesterol, autoimmune diseases, angina, heart attack, cancer, or liver disease. Additionally, we ensured matching for sex, age, and BMI to maintain homogeneity within the study cohorts. Therefore, the rigorous criteria limited the number of participants eligible for inclusion in our study. We recognized that the statement "aiming to enroll 60,000 local participants" could lead to confusion, thus, we removed it. Furthermore, we have augmented our study with an additional 66 participants from Qatar Biobank for external validation, resulting in a total of 133 participants (results section, line 203).

-actin cytoskeleton-regulating proteins are influenced by age and this needs to be discussed

Response: Thank you for your suggestion. We have added the necessary information to clarify the potential effect of age on the identified proteins to the discussion section (line 432).

-Explain how the controls were chosen, because the age was different in the three groups and report how many female patients were in menopause, how many had PCOS which is associated with alterations of the proteome and often hypertension.

Response: Thank you for highlighting the observed age difference between the study groups. Despite our efforts to match for age, a significant difference in age distribution among the groups persisted. However, post hoc analysis (now included in supplementary table S1) revealed no significant age difference between the prehypertension and hypertension groups. This suggests that age does not influence the distinctions between these two groups. Furthermore, it is important to note that hypertension is age-related, and it is expected that control individuals, being healthy, would tend to be younger, as hypertension is less prevalent at a younger age. We have addressed this as a study limitation in the discussion section (line 432).

Regarding menopause and PCOS, we unfortunately do not have available information on these factors. Therefore, we highlighted this as a limitation of our study. We appreciate your feedback and have taken steps to address these limitations in our study (line 428).

-A point to clarify is that the authors exclude patients with obesity but in the table many patients were overweight and obese considering the average BMI reported. Also explain the impact of BMI on the assessment. Report BMI in males and females. Report how many women used contraceptives.

Response: Thank you for your comment. We have addressed your concern by adding BMI data for both males and females to Table 1, as suggested. It's important to note that our study excluded obese individuals with BMI >30, while those classified as overweight (with a BMI between 25 and 29.9) were not excluded. However, we corrected for BMI in our analysis to mitigate any potential confounding effects. As shown in Table 1 no significant difference in BMI across the study groups was observed, suggesting that BMI does not significantly impact our findings.

Furthermore, none of the female participants reported the use of contraceptives

In addition, none of the female participants reported the use of contraceptives.

-An important point to report is whether patients in the three groups were on any diet that could influence proteomics

Response: thank you for your suggestion. We have incorporated the number of patients adhering to a specific diet into Table 1 for the three groups as per your recommendation.

-Explain whether hypertension therapy can influence proteomics and discuss whether each drug may have changed proteomics. Specify whether hypertension therapy and each individual drug can influence the results. This is to explain whether the difference between prehypertension and hypertension is linked to the effect of therapy on proteomics. In figure 4 the standard deviation in cases with hypertension is very high and the authors must discuss whether this is linked to the different types of therapy used

Response: Thank you for your suggestion. We have included the data for patients receiving hypertension treatment in Table 1 as recommended. Chi-square and post hoc analysis indicated no significant difference in the number of patients receiving treatment between prehypertensive and hypertensive groups (p-value=0.41). We have added this information to the results section (line 108) and in supplementary table S1. Furthermore, we conducted additional analysis excluding participants taking medication, which showed that the differences in our proteins of interest between the groups remained significant (analysis is added to the results section, lines 132 and 152, and supplementary Figure S2 and Table S4). These findings suggest that medication use does not significantly influence our proteomic results. We would like to bring to your attention that most patients do not specify the names of their medications, they only indicate the use of anti-hypertension tablets. Therefore, we lack detailed information about individual drug names.

Regarding the high standard deviation observed in Figure 4, we attempted to exclude participants receiving therapy from the box plots. However, we observed minimal change in the error bars, suggesting that medication usage may not be the primary cause of the high standard deviation. If requested by the reviewer, we are prepared to include these plots in the supplementary materials.

Round 2

Reviewer 1 Report

Comments and Suggestions for Authors

Indeed the authors have worked very hard to have a corrected version of their original version in a short time. However, the changes made are not entirely clear:

what does: in the discovery cohort means?

DASH: Dietary Approaches to Stop Hypertension, what consist of?

Authors inserted two rows to indicate hypertensive treatment, and indicated tablets of what composition? it is not clear if this initially group was receiving treatment and after they selected another group of patients with no treatment.

The authors indicated that they reanalyzed their data excluding patients receiving medication. Although it seems difficult to find out hypertensive patients without medication, the authors do not clarify with which group they continued their analysis.

Authors performed a validation analysis using a 25 patients and control cohort, they show a ROC curve and a table with p values and the AUC, however they must show their results  in a clearer manner, maybe pointing out the similarity of their values with the original analysis.

The Discussion section is more complete however it is extremely long the meaning of what should be discussed is lost, it needs to be summarized so that it can be read easily and resume the importance and originality of its study.

Author Response

Thank you for your constructive feedback on our manuscript. We sincerely appreciate the time and effort you dedicated to reviewing our work. Below, we are pleased to address your questions, with the corresponding revisions highlighted in the resubmitted manuscript. We trust that these responses adequately address your concerns and meet your expectations.

Comment 1: what does: in the discovery cohort means?

Response: Thank you for your question. In proteomics studies, there are two groups of participants involved. The "discovery cohort" which is the initial set of individuals used for identifying significant biomarkers. After the identification of significant biomarkers in this group, a second set of participants, the "validation cohort," is used. It consists of independent individuals who were not part of the initial discovery group and used to assess whether the identified biomarkers maintain their significance across a different set of individuals. To clarify this in our manuscript, we have included an explanation of the discovery and validation cohorts in the methods section (refer to lines 421 and 424). We hope that this clarification addresses your question.

Comment 2: DASH: Dietary Approaches to Stop Hypertension, what consist of?

Response: We appreciate your question. According to the American Heart Association, the DASH diet is a healthy eating plan that helps to manage blood pressure and reduce the risk of heart attack, stroke and other health threats. It includes food rich in fruits, vegetables, whole grains, low-fat dairy products, and lean meats, while limits food containing saturated fats, salt, and fatty meats (we have now included this information under Table 1, line 117). We hope this addition addresses your concern.

Comment 3: Authors inserted two rows to indicate hypertensive treatment, and indicated tablets of what composition? it is not clear if this initially group was receiving treatment and after they selected another group of patients with no treatment.

Response:  We apologize for any confusion. Regarding the composition of hypertensive tablets, our available questionnaire data only indicated the use of anti-hypertension tablets without specifying their composition. Therefore, detailed information regarding the medication composition is unavailable.

Regarding the selection of participants, we would like to clarify that the initial analysis encompassed all participants, including those receiving treatment. However, in response to your concern about the potential influence of treatment on our findings, we reanalyzed the data from the same participants, but excluding those on medications (The results of this additional analysis were included in Figure S2 and Table S4) Furthermore, we have now clarified this aspect in the results section (lines 134 and 154) by stating that same participants were used in this reanalysis.

Comment 4: The authors indicated that they reanalyzed their data excluding patients receiving medication. Although it seems difficult to find out hypertensive patients without medication, the authors do not clarify with which group they continued their analysis.

Response: Thank you for your feedback. We would like to note that participants with/without hypertension medication were identified based on the questionnaire response, which asks whether participants were being treated for hypertension or not.

Since the results of the reanalysis, excluding patients taking medication, indicated no significant impact of medication on our findings, these additional results were added in supplementary materials (Figure S2 and Table S4) and the downstream analysis was proceeded with all participants included. We have now clearly stated this information in the results section (line 155).

Comment 5: Authors performed a validation analysis using a 25 patients and control cohort, they show a ROC curve and a table with p values and the AUC, however they must show their results in a clearer manner, maybe pointing out the similarity of their values with the original analysis.

Response: Thank you for your suggestion. We have revised Table 4 to display the values from both the initial analysis and the validation analysis together. We hope this adjustment facilitates the understanding of the presented results.

Comment 6: The Discussion section is more complete however it is extremely long the meaning of what should be discussed is lost, it needs to be summarized so that it can be read easily and resume the importance and originality of its study.

Response: Thank you for your feedback. We have revised the Discussion section, eliminated unnecessary content, and summarized it to ensure conciseness while maintaining the essence of the study's importance.

Reviewer 2 Report

Comments and Suggestions for Authors

the Authors have appropriately ansewered to the comments

Author Response

We would like to extend our heartfelt gratitude for your kind acknowledgment. We are delighted that our responses have met your expectations. Your valuable feedback has greatly contributed to the enhancement of our manuscript.

Round 3

Reviewer 1 Report

Comments and Suggestions for Authors

The authors have attended the recommendations